# Ag(I) ions working as a hole-transfer mediator in photoelectrocatalytic water oxidation on WO$_3$ film

Tae Hwa Jeon[1,7], Damián Monllor–Satoca[1,6,7], Gun–hee Moon[1], Wooyul Kim[2], Hyoung–il Kim [3], Detlef W. Bahnemann [4], Hyunwoong Park [5✉] & Wonyong Choi [1✉]

Ag(I) is commonly employed as an electron scavenger to promote water oxidation. In addition to its straightforward role as an electron acceptor, Ag(I) can also capture holes to generate the high-valent silver species. Herein, we demonstrate photoelectrocatalytic (PEC) water oxidation and concurrent dioxygen evolution by the silver redox cycle where Ag(I) acts as a hole-transfer mediator. Ag(I) enhances the PEC performance of WO$_3$ electrodes at 1.23 V vs. RHE with increasing O$_2$ evolution, while forming Ag(II) complexes (Ag$^{II}$NO$_3^+$). Upon turning off both light and potential bias, the photocurrent immediately drops to zero, whereas O$_2$ evolution continues over ~10 h with gradual bleaching of the colored complexes. This phenomenon is observed neither in the Ag(I)-free PEC reactions nor in the photocatalytic (i.e., bias-free) reactions with Ag(I). This study finds that the role of Ag(I) is not limited as an electron scavenger and calls for more thorough studies on the effect of Ag(I).

[1] Division of Environmental Science and Engineering, Pohang University of Science and Technology (POSTECH), Pohang 37673, Korea. [2] Department of Chemical and Biological Engineering, Sookmyung Women's University, Seoul 04310, Korea. [3] Department of Civil and Environmental Engineering, Yonsei University, Seoul 03722, Korea. [4] "Photocatalysis and Nanotechnology", Institut fuer Technische Chemie, Gottfried Wilhelm Leibniz Universitaet Hannover, Hannover, Germany. [5] School of Energy Engineering, Kyungpook National University, Daegu 41566, Korea. [6] Present address: Department of Analytical and Applied Chemistry, Institut Químic de Sarrià (IQS)—School of Engineering, Universitat Ra-mon Llull, Via Augusta, 390, 08017 Barcelona, Spain. [7] These authors contributed equally: Tae Hwa Jeon, Damián Monllor–Satoca ✉email: hwp@knu.ac.kr; wchoi@postech.edu

The selective use of photogenerated charge carriers to induce either substrate oxidation with valence band (VB) holes or reduction with conduction band (CB) electrons on the surface of an irradiated photocatalyst can be generally achieved by suppressing their mutual recombination[1,2], mainly through the addition of sacrificial scavengers of electrons or holes and the application of an external potential that depletes one of the carriers from the photocatalyst[3]. For scavenging CB electrons, the former case commonly uses Ag(I) ions as an electron scavenger to promote oxidative reaction pathways[4], whereas the latter case applies a potential that is more positive than the onset potential to draw electrons out of the photoelectrode[5].

Regarding the effects of Ag(I), some concerns on its role as a water oxidation promoter through the formation of Ag(II) species (i.e., acting as a hole scavenger) have been raised[6]; yet, to date no direct photo(electro)catalytic evidence has been reported supporting the claims. We have noted a potential role of mediated electrocatalytic oxidation involving Ag(II)/Ag(I) redox couple in photocatalytic (PC) and photoelectrocatalytic (PEC) processes. The mediated electrocatalytic oxidation is based on the electrochemical cycling of highly reactive redox shuttles (e.g., $Ag^{2+}/Ag^+$, $Co^{3+}/Co^{2+}$, $Ce^{4+}/Ce^{3+}$, and $Mn^{3+}/Mn^{2+}$) that can easily oxidize many contaminants in a continuous cycle, yielding minimum byproducts[7,8]. It has been used for the degradation of a variety of recalcitrant organic and inorganic compounds[7–12], as well as for water oxidation as a side process[7–9,12]. Recently, homogeneous Ag(I) complex ions of $AgCl_2^-$ and $AgCl_3^{2-}$ were further utilized for electrochemical oxidation of chloride to chlorine[10]. Ag(II)/Ag(I) couple is the best candidate for the mediated electrocatalytic oxidation, as its redox potential is very positive in acidic media ($E^\circ = +1.98\ V_{SHE}$; Supplementary Table 1)[13]. However, Ag(II) is so reactive that it needs to be stabilized using nitrate[7–9,11,14–16], perchlorate[14,17,18], sulfate[9,16], or phosphate[19] as complexing ligands. Nitrate has been frequently used because of its high solubility, low viscosity, and stability[7]; it generates $Ag^{II}NO_3^+$, a dark brown complex[17,20,21] that can promote homogeneous water oxidation (Eqs. 1–4)[7,11]:

$$Ag^+ \rightarrow Ag^{2+} + e^- \qquad (1)$$

$$Ag^{2+} + NO_3^- \rightarrow Ag^{II}NO_3^+ \qquad (2)$$

$$4Ag^{II}NO_3^+ + 2H_2O \rightarrow 4Ag^+ + O_2 + 4H^+ + 4NO_3^- \qquad (3)$$

$$2H_2O \rightarrow O_2 + 4H^+ + 4e^- \qquad (4)$$

As the stability of the complex is paramount for the continuous operation of the above mechanism, rather high concentrations of nitric acid (1–10 M) have been used to prevent the depletion of Ag(II)[17,21], which is normally generated using electrodes with large overpotentials for water oxidation (Pt, boron-doped diamond, etc.). This limits its practical application, as it requires extreme acidic conditions on the one hand, and high applied potentials on the other.

Here we show an example of successful mediated photoelectrocatalytic oxidation (MPEO) of water using a biased $WO_3$ mesoporous electrode in the presence of $Ag^+$, under the conditions of a mild acidic nitrate solution, zero overpotential, and simulated solar light to photogenerate $Ag^{2+}$ that enhances water oxidation via a reversible homogeneous redox cycle of $Ag^{2+}/Ag^+$. $WO_3$ is a common n-type semiconductor that has been actively investigated for the photooxidation of water, owing to its visible light activity (bandgap ca. 2.6–2.7 eV), high oxidation potential of photogenerated holes, and remarkable stability to photocorrosion in acidic conditions (pH < 4)[22–25]. However, the water oxidation efficiency is limited by its sluggish kinetics that renders a partial oxidation of water to adsorbed peroxo species, eventually causing surface deactivation[26]. The presence of $Ag^+$ (as $AgNO_3$) increases the hole lifetime and hence its reactivity[27], as its reduction potential ($E^\circ = +0.80\ V_{SHE}$)[13] is positive enough to scavenge the photogenerated CB electrons[28]. The potential contribution of $Ag^+$ reaction with holes (i.e., MPEO) can be investigated by employing a biased $WO_3$ photoanode, since the positive potential bias can suppress the electron transfer to $Ag^+$ (retarding recombination with hole), but maximize the chance of the hole transfer to $Ag^+$. The combined use of a hole acceptor (or shuttle) with an irradiated biased electrode has not been attempted until recently[23]. Although the role of $Ag^+$ as an electron scavenger in promoting water photooxidation has been well recognized, its role as a hole-transfer mediator via $Ag^{2+}/Ag^+$ redox shuttle in MPEO of water is newly confirmed in this study.

## Results and discussion

**Photoelectrocatalytic performances of $WO_3$ with Ag(I).** Figure 1a depicts the time-profiled photocurrent and concomitant $O_2$ evolution on an irradiated $WO_3$ electrode in a sealed cell containing aqueous sodium nitrate (0.5 M) at pH 5 in the absence and the presence of $AgNO_3$. The employed $WO_3$ electrode was ~12-μm thick and porous, consisting of micrometric nanostructured aggregates (Supplementary Fig. 1). The applied potential was fixed at $E^\circ_{OER}$ (i.e., 1.23 $V_{RHE}$)[13], ensuring that the photocurrent reached a steady-state value (Supplementary Fig. 2). As shown in Fig. 2, the applied potential of $E^\circ_{OER}$ minimizes silver deposition by effectively scavenging the photogenerated electrons (electron transfer from the CB edge to the substrate; ET2 > electron transfer to Ag(I); ET1) and prevents the back reduction of photogenerated $Ag^{2+}$ (hole transfer to Ag(I, II); HT > recombination between electrons in the FTO and Ag(II, III); RE2). In the absence of $AgNO_3$, the photocurrent showed an initial spike followed by a fast decay, caused by the accumulation of peroxo species on the surface[22,23] and the poor charge carrier transport in such a thick film[5,29]. Upon turning off the light, the photocurrent immediately dropped to zero (Supplementary Fig. 3), and the amount of $O_2$ in the headspace remained the same during the dark period in the absence of $AgNO_3$. The overall Faradaic efficiency (FE) of $O_2$ evolution was estimated to be maximum 40% (for 3 h, Fig. 1a inset). In the presence of $AgNO_3$, the photocurrent increased and reached a steady-state value that was up to >4 times higher than that in its absence. Small photocurrent spikes appeared in the presence of $AgNO_3$ (Supplementary Fig. 4), likely due to the sudden release of oxygen bubbles from the $WO_3$ surface (heterogeneous $O_2$ evolution). The accompanying $O_2$ evolution increased with irradiation time (over four times higher than that in the absence of $AgNO_3$), leading to the FE of ~75% (for 3 h), while the solution acquired a brown color, likely caused by the formation of $Ag^{II}NO_3^+$ complex (see Fig. 1b inset)[20,21,30,31]. The catholyte with a counter electrode also had a dark brown coloration likely due to $NO_x$ formed via nitrate reduction[8,16]. The contact of the nitrogen compounds with the anolyte was prevented using glass frits. It is particularly interesting to find that $O_2$ evolution continued over ~10 h even after turning off both electrical bias and light. The gradual bleaching of the electrolyte was accompanied during the unbiased dark period. The $O_2$ amount evolved during the unbiased dark period was ~6 μmol, leading to the overall FE of ~100% for total evolved $O_2$ of ~25.6 μmol (i.e., 19.6 μmol and 6 μmol in the biased irradiation and unbiased dark periods, respectively). Such $O_2$ evolution during the post-PEC period was not observed in the presence of persulfate ($Na_2S_2O_8$), a well-known electron scavenger (Supplementary Fig. 5), which indicates the unique role of $Ag^+$.

The color change of the anolyte was monitored with a UV–Vis spectroscopy during the PEC $O_2$ evolution (Fig. 1b). The clear

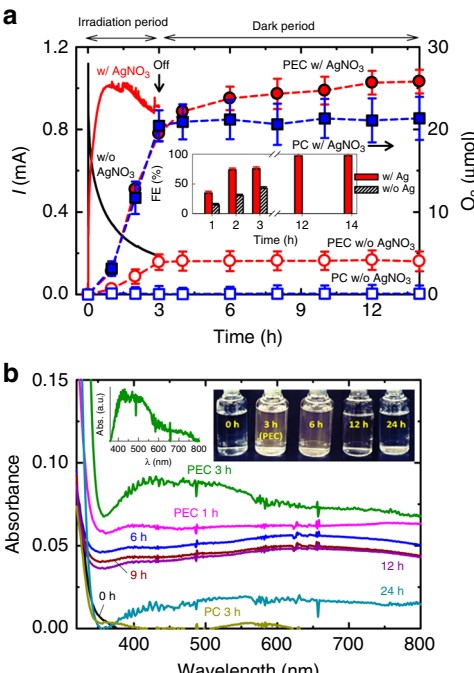

**Fig. 1 Photo(electro)catalytic water oxidation. a** Time-profiled photocurrent (left) and concurrent $O_2$ evolution (right) on a $WO_3$ electrode biased at + 0.74 V vs. Ag/AgCl (i.e., 1.23 V vs. RHE) upon 3 h continuous irradiation in a sealed cell, without (w/o) and with (w/) 50 mM $AgNO_3$. After turning both potential bias and light off, the amount of $O_2$ in the headspace was continuously recorded. For comparison, the photocatalytic (PC) $O_2$ evolution tests with a bias-free $WO_3$ film were performed in the absence and presence of 50 mM $AgNO_3$. Electrolyte: Ar-purged 0.5 M $NaNO_3$, buffered at pH 5.0. Electrode area: 4 $cm^2$. Irradiation: AM 1.5 G (300 mW $cm^{-2}$). Inset shows changes in Faradaic efficiencies (FE) of $O_2$ evolution with time. **b** UV–Vis absorption spectra of electrolytes (0.5 M $NaNO_3$) with $AgNO_3$ (50 mM) used for PEC $O_2$ evolution tests using $WO_3$ electrode. Before the PEC test, the electrolyte did not show any significant absorbance; during the PEC period of 3 h, the transparent electrolyte gradually turned brown; after turning off the bias and light, then the brown electrolyte was gradually bleached (6–24 h). The insets show (left) the enlarged absorption spectrum of the PEC-3 h sample (spectrum subtracted by the absorbance at 800 nm) and (right) the photo images of the electrolytes. For comparison, the absorption spectrum of the electrolyte used for PC (for 3 h) is shown together. Source data are provided as a Source Data file.

anolyte (i.e., 0.5 M $NaNO_3$ + 50 mM $AgNO_3$) gradually turned brown during the PEC period of 3 h while a broad absorption band appeared in the wavelength range of 400−600 nm. Although the band intensity was significantly weak with strong background absorption, the spectrum subtracted by the background absorbance (Abs@$\lambda$ = 800 nm) (Fig. 1b inset) was similar to that obtained under highly acidic conditions (1.5–6 M $HNO_3$)[17]. This similarity suggests that the observed band could be attributed to the internal ligand field d → d transitions of Ag(II) complexes. The structure of the brown complex needs to be further studied in detail. Upon turning off the bias and light, the brown color gradually disappeared over 24 h. The observations that the $O_2$ evolution concurred with the brown coloration and that both the $O_2$ evolution and brown color persisted for many hours (with gradual decay) even after the turning off the light, suggest that the brown complex can be responsible for the $O_2$ evolution in the post-PEC period. It further implies the existence of an alternative

pathway for the $O_2$ evolution mediated by the brown complex in the homogeneous solution phase (homogeneous $O_2$ evolution).

**Confirmation of high-valent silver species.** To identify the oxidation state of the Ag species in the complex, the EPR analysis was performed on the complex adsorbed on silica gel after the PEC reaction (Ag(PEC)) (Fig. 3). The complex exhibited the strong rhombic symmetric spectrum with two distinct g values of 2.80 ($g_{zz}$) and 2.16 ($g_{yy}$). This spectrum was similar to that of AgO power. The absence of the g value close to 1.52 ($g_{xx}$) in the Ag(PEC) sample (Supplementary Fig. 6) was attributed likely to different structures of Ag(II) (complex vs. oxide). For further comparison, the EPR spectrum of $Ag_2O$ powder was examined as well; it was axially symmetric with two distinct g values of 3.43 ($g_{zz}$) and 2.04 ($g_{yy} = g_{xx}$) and clearly different from that of the Ag(PEC) sample (Supplementary Fig. 6). No complex was created in the PC reaction; hence the Ag(PC) sample (i.e., mixed with silica gel and dried) did not show any specific peak (Fig. 3) and it was similar to a control (silica gel only). Considering a negative shift in the g value for oxidized transition metal oxides/complexes ($Ag_2O$ vs. Ag(PEC))[32,33] and the similarity of the spectra between AgO and Ag(PEC), the brown complex should be primarily composed of Ag(II).

The Ag(II) in the brown complex was further quantified using Fe(II) as a reducing agent according to a well-established chemical redox reaction[34]. The reduction of Ag(II) to Ag(I) ($E°$ ($Ag^{2+/+}$) = 1.980 V) can lead to a fast oxidation of Fe(II) to Fe (III) ($E°(Fe^{3+/2+})$ = 0.771 V), whereas a further reduction of Ag (I) to Ag(0) by Fe(II) should be slow due to similar reduction potentials of Ag(I)/Ag(0) ($E°(Ag^{2+/+})$ = 0.7996 V) and Fe(III)/Fe (II) redox couples. No production of Fe(III) in aqueous $Ag^INO_3$ solutions verifies the insignificant redox reaction in the latter (Supplementary Fig. 7). During the PEC reaction at $E°_{OER}$, the amount of Ag(II) linearly increased with irradiation time, whereas there was no Ag(II) during the PC reaction (Fig. 4; Supplementary Fig. 8). This indicates that the Ag(II) production essentially required irradiation as well as a potential bias. In other words, the $O_2$ evolution in the PEC reaction accompanies the oxidation of Ag(I) to Ag(II) (i.e., hole-mediated OER), whereas Ag(I) is used only as an electron acceptor in the PC reaction. During the post-PEC period (3–24 h), the amount of Ag(II) produced in the PEC period gradually decreased by 90% at 12 h and ~95% at 24 h. The FE of Ag(II) production in the PEC period was ~22%; hence the overall FE became ~97%, including $O_2$ evolution (FE ~75%; see Fig. 1a). The simultaneous production of Ag(II) in the PEC reaction was observed at lower and higher potentials than $E°_{OER}$ (1.04 and 1.79 V, respectively; see Supplementary Fig. 9a, b). Both FEs were estimated to be ~21 and 29%, respectively, leading to the overall FEs of ~95%, including $O_2$ evolution (Fig. 4 inset).

All experimental evidences indicate the presence of the water oxidation pathway that should be mediated by the in situ formed Ag(II) (Eq. 5; Fig. 5):

$$Ag^+ + h^+ \rightarrow Ag^{2+} \tag{5}$$

Upon stabilization by nitrate complexation, Ag(II) can oxidize water in a homogeneous process (Eqs. 2, 3). Such process should be possible only under positive bias, as the applied positive potential effectively extracts the photogenerated electrons from $WO_3$ particles. However, in the absence of the positive bias, both charge recombination and silver deposition (Eq. 6) can occur predominantly.

$$Ag^+ + e^- \rightarrow Ag^0 \tag{6}$$

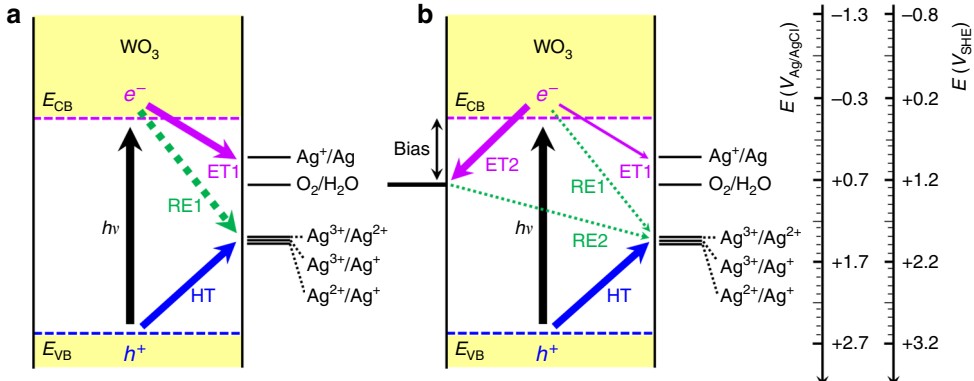

**Fig. 2 Photoinduced charge transfers occurring on the irradiated WO₃.** A simplified band diagram of WO₃ coupled with the electrolyte redox levels and the photoinduced charge transfers occurring on the irradiated WO₃ electrode in the presence of AgNO₃ (pH 5), (**a**) under open-circuit (i.e., bias-free) condition and (**b**) biased at +0.74 V vs. Ag/AgCl (equivalent to 1.23 V vs. RHE). They are denoted as photocatalytic (PC) and photoelectrocatalytic (PEC) processes, respectively, throughout the text. $E_{CB}$ and $E_{VB}$ represent conduction band edge (+0.03 V) and valence band edge (+2.73 V), respectively. Helmholtz layer potential drop at the interface and the redox energy level broadening caused by thermal fluctuation (i.e., reorganization energy) are not represented. The effect of nitrate complexation on the $Ag^{2+}/Ag^{+}$ redox potential is not considered. Colored arrows depict the possible charge transfer steps: (ET1) electron transfer to Ag(I); (ET2) electron transfer from the CB edge to substrate (FTO); (HT) hole transfer to Ag(I, II); (RE1) recombination between electrons in the CB edge and Ag(II, III); (RE2) recombination between electrons in the FTO and Ag(II, III). Thicker arrows indicate more dominant paths.

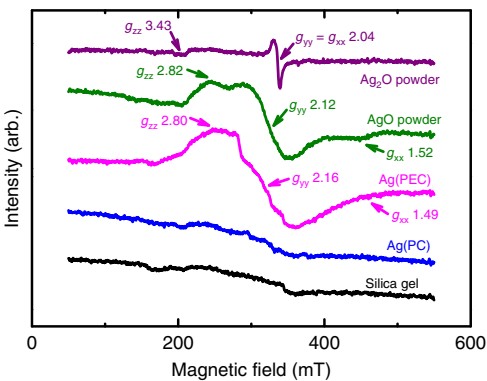

**Fig. 3 EPR spectra of high-valent silver complexes.** Electron paramagnetic resonance (EPR) spectra of silver-containing aliquots after PEC and PC reactions for 3 h (denoted as Ag(PEC) and Ag(PC), respectively). The aliquots were adsorbed onto silica gel. See Fig. 1a for the detailed experiments. For comparison, the EPR spectra of commercial silver powders (AgO and Ag₂O) and silica gel are shown. See Supplementary Fig. 6 for more detailed comparison of the EPR spectra. Source data are provided as a Source Data file.

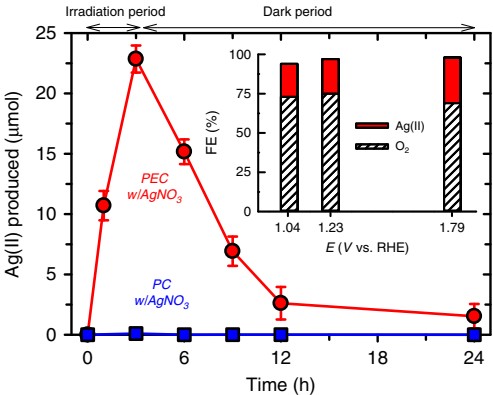

**Fig. 4 Quantification of Ag(II) species.** Ag(II) productions during the irradiation (PC and PEC) periods (0–3 h) and subsequent dark periods (3–24 h) of WO₃ in the presence of AgNO₃ (50 mM). The experimental conditions of PC (i.e., open circuit) and PEC (held at 1.23 V vs. RHE) reactions with AgNO₃ were the same as those in Fig. 1a. No Ag(II) was found in the PC process. The amounts of Ag(II) were estimated with the reaction of Fe(II). See text for more detailed analytical method. The inset shows Faradaic efficiencies (FE) for O₂ evolution reaction (OER) and Ag(II) production at various applied potentials. Source data are provided as a Source Data file.

To examine this possibility, the same experiment was repeated under irradiation without any bias (i.e., PC condition, virtually equivalent to open-circuit condition) and AgNO₃. Not surprisingly, O₂ was not evolved due to the predominant charge recombination on bare WO₃ particles (Fig. 1a). In the presence of AgNO₃, PC process evolved O₂ at a rate of ~7 μmol h⁻¹, which was nearly the same as that of the PEC process with AgNO₃ and yet approximately fivefold that of the PEC without AgNO₃. The latter fact (PC with AgNO₃ » PEC without AgNO₃ for O₂ evolution) indicates that the electron-scavenging efficiency of Ag⁺ in the PC process is higher than that of the potential bias in the PEC process. However, such the electron-scavenging role of Ag⁺ found in the PC process should be limited in the PEC process where the electron transfer to Ag⁺ (Eq. 6) is hindered under the positively biased condition. Nevertheless, the fact that the O₂ evolution in the PC system with AgNO₃ is similar to that

in the PEC system with AgNO₃ implies a different role of Ag⁺ in the PEC process. After the PC reaction for 3 h, as expected, Ag deposition onto WO₃ occurred via the reduction of Ag(I) by photogenerated electrons (Eq. 6). Once deposited, however, silver particles inhibited the photocurrent generation on WO₃ electrode biased at $E°_{OER}$ in bare electrolyte (Supplementary Figs. 3 and 9c) due to electron accumulation on the Ag deposits through the Ag/ WO₃ heterojunction[35]. Apparently, the surface plasmon resonance of deposited silver particles[36,37] did not contribute to photocurrent. In addition, the unbiased WO₃ film acquired a bright white metallic luster due to the formation of deposited silver clusters (Supplementary Fig. 10).

The Ag-deposited WO₃ films that were obtained after the PC and PEC experiments of Fig. 1a (denoted as Ag(PC)/WO₃ and Ag

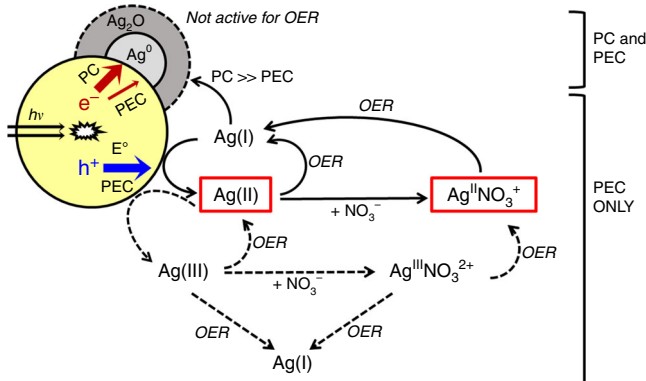

**Fig. 5 Silver(I)-mediated oxygen evolution pathway.** Schematic illustration of Ag(I)-mediated oxygen evolution reaction (OER) in photocatalysis (PC) and photoelectrocatalysis (PEC, biased at $E° = 1.23$ V vs. RHE) with $WO_3$ in aqueous solutions of $Ag^I NO_3$. Some silver species (i.e., Ag(II), $Ag^{II}NO_3^+$, $Ag^0$, and $Ag_2O$) were identified directly and indirectly (solid arrows), whereas trivalent silver species (Ag(III) and $Ag^{III}NO_3^{2+}$) were speculated to contribute to the OER (dashed arrows).

(PEC)/$WO_3$, respectively) were reused for PEC $O_2$ evolution tests in Ag(I)-free solutions. Both electrodes exhibited the similar levels of photocurrents and amounts of evolved $O_2$ (3.5–4 μmol for 3 h; FE < 20%; see Supplementary Fig. 9c). The photocurrent shapes resembled that of the PEC with $AgNO_3$ case in Fig. 1a (i.e., $WO_3$ with $AgNO_3$). However, the PEC $O_2$ evolution activities of the Ag-loaded electrodes were much lower than the PEC with $AgNO_3$ case and comparable with the PEC without $AgNO_3$ case (Fig. 1a). In addition, both Ag-loaded electrodes exhibited no sign of $O_2$ evolution in the post-PEC periods, unlike the case of Fig. 1a. Therefore, Ag nanoparticles loaded on $WO_3$ electrode (via either PC or PEC process) can partially inhibit the charge recombination, but its role in facilitating $O_2$ evolution is limited. This indicates that the markedly enhanced PEC $O_2$ evolution observed in the case of $WO_3$ with $AgNO_3$ in Fig. 1a should not be attributed to the role of in situ photodeposited Ag nanoparticles.

**Structure and characterization of Ag-deposited $WO_3$.** The XRD measurements of the Ag(PEC)/$WO_3$ and Ag(PC)/$WO_3$ samples revealed that the Ag deposits were crystalline (Fig. 6a). Their main components were cubic $Ag^0$ and $Ag_2O$, although traces of AgO ($2\theta = 37.7°$) could also be detected[38]; no traces of crystallized $Ag_x WO_{3-x}$, $Ag_2O_3$, $Ag_3O_4$, or $Ag_7O_8NO_3$ phases were observed. After deconvolution of the most intense XRD Ag (111) and $Ag_2O$ (200) peaks (Fig. 6a inset)[39], the primary particle sizes of Ag and $Ag_2O$ were estimated to be in the range 110 nm (Ag) to 170 nm ($Ag_2O$) (Supplementary Table 2) using Scherrer equation[40]. The deposition of the Ag and $Ag_2O$ particles on $WO_3$ requires the pre-adsorption of Ag(I); $WO_3$ surface is negatively charged at pH 5 due to a low point-of-zero charge ($pH_{pzc} = 0.3–0.5$)[41], which facilitates $Ag^+$ adsorption and subsequent reductive deposition (Eq. 6). $Ag_2O$ could be formed from the reaction between Ag(I) and surface oxygen species (e.g., Eq. 7) and then reduced to Ag (Eq. 8)[42]. Furthermore, Wang et al.[43] recently suggested that $Ag_2O$ (bandgap 1.2 eV) can be photo-reduced to Ag under visible light irradiation (Eq. 9).

$$2Ag^+ + O^{2-} \rightarrow Ag_2O \tag{7}$$

$$Ag_2O + 2H^+ + e^- \rightarrow 2Ag + H_2O \tag{8}$$

$$Ag_2O + h\nu \rightarrow 2Ag + 1/2 O_2 \tag{9}$$

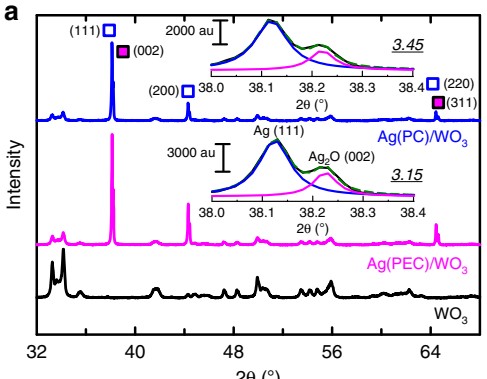

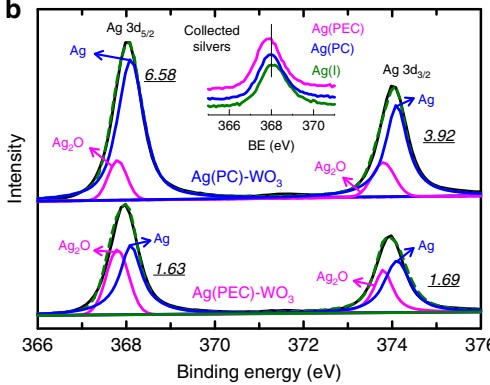

**Fig. 6 Structure and characterization of Ag-deposited $WO_3$. a** XRD patterns and **b** Ag 3d XPS spectra of bare $WO_3$ (black), Ag(PEC)/$WO_3$ (red), and Ag(PC)/$WO_3$ (blue) samples. In inset in (**a**) shows deconvoluted Ag (111) and $Ag_2O$ (002) peaks. Diffraction peaks: (i) Ag (void blue squares): $2\theta = 38.2°$ (111), 44.3° (200), 64.5° (220), (ii) $Ag_2O$ (red filled squares): $2\theta = 38.2°$ (002), 65.5° (311). XPS peaks ($3d_{5/2}$ and $3d_{3/2}$): 368.08 and 374.08 eV (Ag); 367.78 and 373.78 eV ($Ag_2O$), respectively. The underlined numbers in (**a**) and (**b**) are the ratios of Ag/$Ag_2O$. The inset in (**b**) shows the Ag $3d_{5/2}$ bands for the solution samples collected via adsorption onto silica gel before (i.e., Ag(I)) and after PEC and PC reactions. See Supplementary Fig. 14 for the full spectra. Source data are provided as a Source Data file.

Ag(I) can also trap the photogenerated oxygen species as a form of $Ag_2O$, enhancing water photooxidation by its decomposition to Ag and $O_2$.

XPS measurements for both samples showed the presence of Ag 3d-doublets and their deconvolution further revealed the presence of Ag and $Ag_2O$ (Fig. 6b). In the Ag(PEC)/$WO_3$ sample, the Ag/$Ag_2O$ ratios for the $3d_{5/2}$ and $3d_{3/2}$ peaks were estimated to be 1.63 and 1.69, respectively; however, in the Ag (PC)/$WO_3$ sample, they were 6.58 and 3.92, respectively. This analysis suggests that most of Ag(I) was photocatalytically reduced to $Ag^0$ (~80%) in the absence of a potential bias, whereas ~60% of Ag(I) remained as $Ag_2O$ (i.e., the fraction of $Ag^0$ ~40%) under a potential bias. On the other hand, W4f and O1s peaks shifted to lower binding energies (Supplementary Fig. 11), indicating the formation of a new phase on the surface of the parent $WO_3$[44]. These XPS and XRD results suggest that a part of the deposits might have a core–shell structure with $Ag_2O$ on the surface and Ag in the inner core (Ag@$Ag_2O$), in accordance with Eqs. 6–9.

The HR-TEM analysis of both samples confirmed the core–shell structure of Ag@$Ag_2O$. As shown in Fig. 7a, b, the

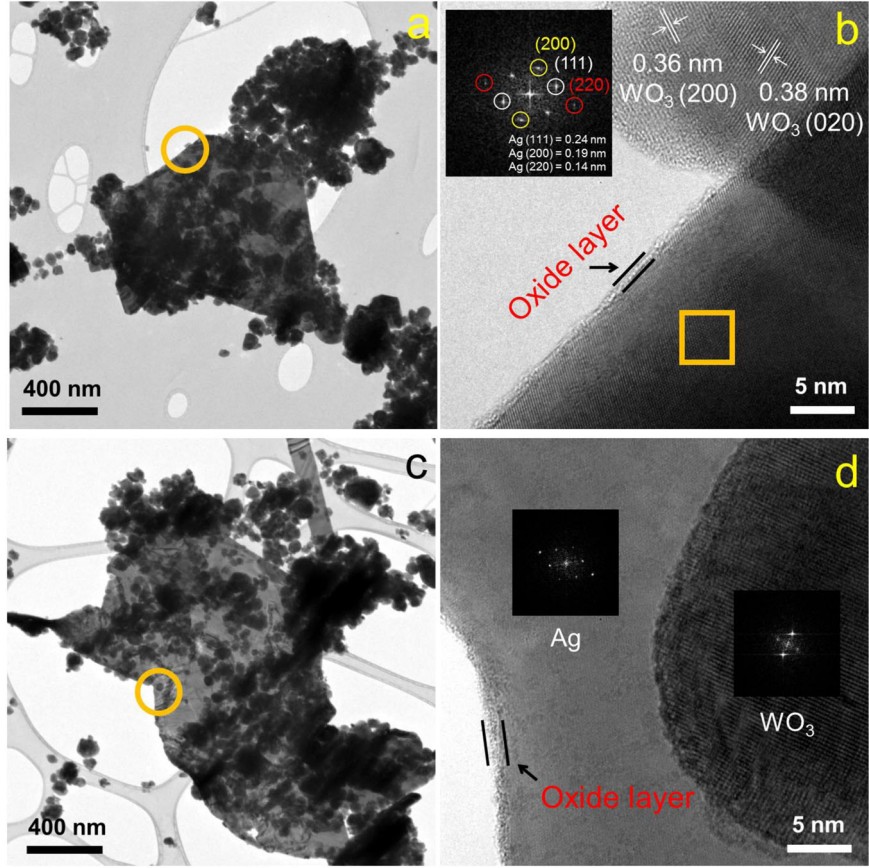

**Fig. 7 Morphology of Ag-deposited WO₃.** TEM images of (**a**, **b**) Ag(PEC)/WO₃ and (**c**, **d**) Ag(PC)/WO₃ particles. The selected areas (orange squares) in the plates were analyzed for the FFT diffraction patterns.

Ag deposits showed the plate configuration of ~1 μm size, which was in direct contact with WO₃ with lattice fringe spacing of 0.36 nm (200) and 0.38 nm (020). The FFT diffraction patterns of the plate interior (bulk) confirmed $Ag^0$ (111, 200, and 220), whereas its edge was covered with an amorphous oxide layer of ~1 nm (Supplementary Fig. 12). The Ag deposits in the PC sample showed the same core–shell structure (Fig. 7c, d; Supplementary Fig. 13). It is noteworthy that the similar XPS surface analysis of Ag(PC)/WO₃ and Ag(PEC)/WO₃ samples indicates the applied bias in the latter was not an optimum condition to prevent the formation of silver deposits. On the other hand, the brown complexes showed the shifted Ag 3d peaks, whereas the binding energy of the silver species after PC reaction was the same as that of Ag(I) (Fig. 6b inset; Supplementary Fig. 14). In contrast to the deposited ones, silver species in the brown complexes existed as a more oxidized state than $Ag^+$, which is consistent with the result of the EPR spectrum on the complexes (Fig. 4).

Regardless of the states of the Ag deposits, the pathway in Eq. 9 does not justify such an increased oxygen production as confirmed in Supplementary Fig. 9c. In addition, no suspended particles were observed, precluding the formation of solid $Ag_2O$/ AgO in the electrolyte. In this regard, the water oxidation should proceed both heterogeneously (through WO₃ with or without $Ag_2O$) and homogeneously (through dissolved $Ag^{II}NO_3^+$). In the former, silver deposits ($Ag_2O$) can work as a water oxidation site or catalyst; however, their role in water oxidation (i.e., Eq. 9) is limited as the reuse tests of the Ag-loaded WO₃ electrodes revealed (Supplementary Fig. 9c). Instead, the water oxidation is presumed to occur predominantly on Ag-free WO₃ surface.

In the absence of potential bias (i.e., photocatalysis), the heterogeneous water oxidation is predominant, whereas the homogeneous pathway should be limited because the electron transfer to the interfacial Ag(I) is significantly faster than the hole transfer, depleting nearby Ag(I) available for holes (Fig. 2). A potential bias can retard the Ag(I) reduction by abstracting electrons from WO₃ to the FTO substrate, increasing the Ag(I) availability for reaction with holes on the surface of WO₃. In addition, nitrate did not show any parallel hole-mediated oxidation[23], but it can be photoreduced as the CB edge has a potential negative enough (Supplementary Table 1). Nevertheless, any nitrate effect can be ruled out as no significant change (i.e., abnormal current or electrolyte color) was observed neither in the dark nor under irradiation in the voltammetric potential range explored (Supplementary Fig. 2). This is consistent with the reported slow nitrate photoreduction using unmodified semiconductors[45,46].

The homogeneous oxygen evolution can proceed through the fast dismutation of Ag(II) (e.g., $Ag^{II}NO_3^+$; Eq. 10)[20], followed by water oxidation by Ag(III) (e.g., $Ag^{III}NO_3^{2+}$) and concurrent Ag (I) regeneration (Eq. 11; Fig. 5)[17,20,23]:

$$2Ag^{II}NO_3^+ \rightarrow Ag^+ + Ag^{III}NO_3^{2+} + NO_3^- \qquad (10)$$

$$Ag^{III}NO_3^{2+} + H_2O \rightarrow Ag^+ + 1/2O_2 + 2H^+ + NO_3^- \qquad (11)$$

$$Ag^{II}NO_3^+ + h^+ \rightarrow Ag^{III}NO_3^{2+} \qquad (12)$$

$$2Ag^{III}NO_3^{2+} + H_2O \rightarrow 2Ag^{II}NO_3^+ + 1/2O_2 + 2H^+ \qquad (13)$$

Although the trivalent Ag species (Ag(III)) was not confirmed in this study, similar reduction potentials of Ag(II) and Ag(III) ($E°$ (Ag$^{3+/2+}$) = 1.8 V; $E°$ (Ag$^{3+/+}$) = 1.9; $E°$ (Ag$^{2+/+}$) = 1.98 V) open up the possibility of the direct Ag(III) formation. For example, Ag(II) can be further oxidized to Ag(III) by the VB holes (Eq. 12) and then the Ag(III) is reduced to Ag(II) and Ag(I) while evolving $O_2$ (Eqs. 13 and 11, respectively) during the PEC period. A further study is needed.

In this contribution, we demonstrated water photooxidation with employing a positively biased $WO_3$ mesoporous electrode in the presence of $AgNO_3$ acting as an efficient hole scavenger through a regenerative Ag(II)/Ag(I) redox cycle. Ag(I) enhanced the PEC performance of $WO_3$ electrodes at 1.23 V vs. RHE with increasing $O_2$ evolution, while forming Ag(II) complexes (Ag$^{II}$NO$_3^+$). Upon turning off both light and potential bias, the photocurrent immediately dropped to zero, whereas $O_2$ evolution continued over ~10 h with gradual bleaching of the colored Ag$^{II}$NO$_3^+$ complexes. The observed behaviors of the PEC water oxidation with $AgNO_3$ were contrary to the commonly accepted role of Ag(I) as an electron scavenger in conventional photo-catalysis. This is a practical example of mediated PEC oxidation of water that operates with mild acidic nitrate conditions in contrast with the highly acidic and concentrated conditions employed in conventional mediated electrochemical oxidation, the thermodynamic potential for water oxidation without the need of overpotential, and a visible-active photocatalyst under (simulated) solar light irradiation. The photooxidation of water proceeded through parallel mechanistic paths: a homogeneous one mediated by complexed high-valent silver species and a heterogeneous one. Further studies should attempt to elucidate the detailed mechanism of the oxygen evolution reaction, optimize all experimental conditions to obtain higher efficiencies by minimizing Ag deposition, and explore the use of other semiconductors, redox couples, and background electrolytes.

## Methods

**Materials and electrodes**. All chemicals were used as received without further purification. $NaNO_3$ (Aldrich, 99.99 + %) and $AgNO_3$ (Sigma-Aldrich, ACS Reagent, 99 + %) were used for preparing the electrolyte. Other chemicals used were $HClO_4$ (Sigma-Aldrich, ACS Reagent, 70%), NaOH (Fluka, Standard Solution, 8 M), $Na_2S_2O_8$ (Sigma-Aldrich), and polyethylene glycol #20,000 (Samchun Chemicals). Solutions were prepared using ultrapure distilled water (18.3 MΩ cm, Barnstead EASYpure RO system). Tungsten trioxide (Aldrich, tungstic anhydride, average particle size < 100 nm) was used as a photocatalyst.

The electrodes were prepared onto fluorine-doped tin oxide substrates (FTO, Pilkington, 15 Ω square$^{-1}$). $WO_3$ nanopowder of 3.2 g was mixed with 2 mL solution of polyethylene glycol (15 g) and water (15 mL). The mixture was ground for 2–3 min until a homogeneous sticky dispersion was obtained. Then some drops of the slurry were spread by the doctor blade method over the FTO, defining an approximate area of 4.0 cm$^2$. Finally, the electrode was dried in open air, and subsequently heated at 450 °C for 1 h for sintering the nanoparticles and burning out the organic binder. The prepared electrodes were stable enough for repeated electrochemical measurements in this work. The average deposited mass of $WO_3$ was 350(± 30) mg. The average electrode thickness was ~12 μm (Supplementary Fig. 1).

**Photoelectrocatalytic measurement and analysis**. Photoelectrocatalytic (PEC) experiments were performed using potentiostat–galvanostat (Gamry Instruments Reference 600) that was connected to a three-electrode electrochemical Pyrex cell. The cell consisted of three compartments, each of which held the working, reference, and counter electrode, respectively. The working electrode compartment had a total volume of 55 mL. All compartments were separated by glass frits to inhibit the direct contact between the reaction products generated on the working electrode (e.g., $O_2$, $H_2O_2$, and Ag$^{II}$NO$_3^+$) and the counter electrode (e.g., $H_2$, NO$_x$, and Ag). The working electrode was a mesoporous $WO_3$/FTO thin film (geometric exposed area, ~4 cm$^2$), attached from the outside of the cell with the $WO_3$ side in contact with the electrolyte. A Pt wire was used as a counter electrode. Potentials were measured and referred against an Ag/AgCl/KCl (sat'd) electrode ($E°$ = + 0.197 V$_{SHE}$). The electrolyte was 0.5 M $NaNO_3$ solution, adjusted to pH 5.0 with concentrated $HClO_4$ or NaOH solution, and purged with Ar gas for 1 h prior to any measurement. If necessary, purging was continuously kept above the electrolyte (headspace), and the electrolyte remained unstirred (stagnant condition). The

concentration of $AgNO_3$ was adjusted to 50 mM using 0.5 M stock solution. For comparison, $Na_2S_2O_8$ (50 mM) was used as an alternative electron scavenger. The light source was an AM 1.5 G solar simulator (Abet Technologies LS1-10500), with an in-built ozone-free 150-W Xe arc lamp (model LS150). The electrode was irradiated from the back side (i.e., FTO → $WO_3$ → solution). The incident light intensity was measured with a thermopile head (Newport 818P-001-12) connected to an optical power meter (Newport 1918-R), yielding 300(± 15) mW cm$^{-2}$ (ca. 3 suns). For comparison, the aforementioned experiments were performed without any potential bias (i.e., open circuit) to drive photocatalytic (PC) reactions. All experiments were performed at room temperature.

The amount of photogenerated $O_2$ in the reactor headspace (4.56 ± 0.01 mL) was analyzed using a gas chromatograph (GC, HP6890A) equipped with a thermal conductivity detector (TCD) and a 5-Å molecular sieve column. Ultrahigh purity argon (Linde Korea, 99.999%) was used as carrier gas. In total, 100 μL of gas samples were intermittently withdrawn from the working electrode headspace with a gastight glass syringe (Hamilton 81030). All cell compartments were thoroughly sealed with rubber septa, glycerine and parafilm, to prevent any gas leakage. Prior to any measurement, the electrolyte was Ar-purged for 1 h. The Faradaic efficiency (FE) for $O_2$ evolution was estimated by the following equation:

$$FE = 4Fn/Q_{ph} \times 100\%, \quad (14)$$

where $F$ is the Faraday constant, $n$ is the measured amount of evolved $O_2$, and $Q_{ph}$ is the integrated photocharge. For the PEC tests, unless otherwise specified, the electrode was continuously held at + 0.74 V vs. Ag/AgCl (equivalent to $E°_{OER}$ = 1.23 V$_{RHE}$ at pH 5) under irradiation for 3 h, and then both potential bias and irradiation were off. In the PC tests, $WO_3$ alone was irradiated without the bias for 3 h. The addition of $AgNO_3$ was performed with the unsealed cell, under vigorous stirring and Ar purging. The amount of photogenerated oxygen was subtracted by an initial small background $O_2$ level (0.27 ± 0.09 μmol) that cannot be removed by Ar purging. The initial background $O_2$ level did not increase over 12 h in the dark control condition, which indicates a negligible air leaking. All experiments were performed more than twice at room temperature.

The amount of produced Ag(II) in the solution was estimated with the oxidation of Fe(II) to Fe(III)[34], and change in concentration of Fe(II) was analyzed by the Ferrozine method. In brief, 0.01 M Ferrozine (3-(2-pyridyl)-5,6-diphenyl-1,2,4-triazine-p,p'-disulfonic acid monosodium salt hydrate; Sigma-Aldrich, 97%), 100 ppm Mohr's salt ((NH$_4$)$_2$Fe(SO$_4$)$_2$·6H$_2$O; Sigma-Aldrich, 99%), and 7 M ammonium acetate (CH$_3$CO$_2$NH$_4$; Sigma-Aldrich, 97%) buffer (pH 9.5 adjusted by concentrated NH$_4$OH) solutions were prepared in advance. Aliquots (0.1 mL) were intermittently withdrawn from the working electrode compartment during irradiated and dark periods, and then diluted with deionized water (0.9 mL). The diluted samples (0.3 mL) were mixed with the Mohr's salt (0.3 mL), Ferrozine (100 μL), buffer solution (15 μL), and deionized water (2.4 mL). The mixed solutions were kept under vigorous stirring for 10 min before measuring the absorbance at λ = 560 nm using a UV–Vis absorption spectrophotometer (Libra S22, Biochrom).

**Surface characterization**. The light absorption of the photogenerated Ag$^{II}$NO$_3^+$ complex was monitored using an Agilent 8453 spectrophotometer. The cell optical path length was 1 cm, and its sample volume was 3 mL. Samples were prepared by intermittently extracting 1 mL aliquots from the electrolyte and diluting them to 3 mL with fresh background electrolyte (0.5 M $NaNO_3$). The sample extraction was intermittently performed during the irradiation and the dark period under Ar purging. All spectra were referenced against the background electrolyte without $AgNO_3$. The surface morphologies of the bare $WO_3$ and silver-deposited $WO_3$ samples were analyzed using high-resolution field-emission scanning electron microscopy (FE-SEM, JOEL JSM-7800F PRIME) with dual-energy dispersive X-ray spectroscopy (EDS) and high-resolution field-emission transmission electron microscopy (HR-FE-TEM, JOEL JEM-2200FS) with image Cs-corrected at National Institute for Nanomaterials Technology (Pohang, Korea). X-ray diffraction (XRD) patterns of the samples were measured using Cu Kα radiation (RIGAKU D MAX 2500). X-ray photoelectron spectroscopy (XPS) was analyzed using monochromated Al Kα radiation as an X-ray source (1486.6 eV) at Korea Basic Science Institute (Busan Center, Korea). Electron paramagnetic resonance (EPR) spectroscopy was analyzed using Bruker EMX/Plus spectrometer equipped with a dual-mode cavity (ER4116DM) at Korea Basic Science Institute (Western Seoul Center, Korea). EPR data were obtained under the following conditions: microwave frequency of 9.64 GHz, a modulation amplitude of 10 G, modulation frequency of 100 kHz, microwave power of 0.92 mW, and a temperature of 298 K.

## Data availability

All relevant data are available within the Article, Supplementary Information, Source Data file or available from the corresponding authors upon reasonable request. The source data underlying Figs. 1, 3, 4, and 6, and Supplementary Figs. 2, 3, 4, 5, 6a–c, 7, 8a–d, 9a–c, and 14 are provided as a Source Data file.

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

## Acknowledgements

This work was supported by the Global Research Laboratory (GRL) Program (No. NRF-2014K1A2041044) and "Next Generation Carbon Upcycling Project" (Project No. 2017M1A2A2046736), which were funded by the Korea Government (Ministry of Science and ICT) through the National Research Foundation (NRF). H.P. is grateful to the National Research Foundation of Korea (2019R1A2C2002602 and 2019M1A2A2065616).

## Author contributions

T.H.J., D.M.-S., H.P., and W.C. designed the materials and carried out the experiments and data analysis. G.-h.M., W.K., H.-i.K., and D.W.B. contributed to discussion on the data analysis. All the authors participate in the discussion on the experimental results. T. H.J., D.M.-S., H.P., and W.C. wrote the paper.

## Competing interests

The authors declare no competing interests.
