## [Peer Review File · Nature Communications]

Reviewers' comments:

Reviewer #1 (Remarks to the Author):

The authors present a carefully executed study on the role of Ag ions as hole-transfer mediator during photoelectrochemical water oxidation. This is a very important study that will influence the behaviour of groups developing, testing and carrying out mechanistic studies of, photoelectrodes and photocatalysts. Ag⁺ is a common electron scavenger and here it is clearly shown that in addition to its role as SEA during PC it can also enable improved photocurrents and FE for water oxidation. This has 2 main implications (1) many mechanistic studies need to be carefully re-evaluated and (2) it also offers a simple way to potentially improve the efficiency of photoanodes such as WO₃ which have historically shown low FE's for O₂ production. I strongly recommend publication with only a few points to change/clarify.

1. In the caption of scheme 1 it is stated that band bending is precluded as the WO₃ size is < 100 nm. Has this actually been checked? depletion widths are typically < 50 nm. This is important as throughout the manuscript the authors state that the applied bias during PEC experiments lowers Ag⁺ reduction and enhances Ag⁺ oxidation by holes. If the WO₃ is fully depleted what is the mechanism by which holes are swept to the surface but electrons moved into the bulk?
2. Line 172/173. A O₂ leak is mentioned and it is stated that a constant value of 0.27 μmol is subtracted off. If O₂ is leaking into the cell I would expect the amount of leaked in air would increase with time. Is this value actually a leak rate e.g. $\mu\text{mol hr}^{-1}$?
3. Figure 1. The insets are very small and hard to see - suggest making these separate figures. The FE is an important data set (presumably this should also have error bars)
4. Figure 1. I cant see error bars on the O₂ levels w/o Ag⁺

Reviewer #2 (Remarks to the Author):

This paper presents a solid piece of work showing that Ag(II) can be generated in photoelectrocatalytic experiments using WO₃ on an electrode. The work is well carried out and well written. I have no specific criticisms or remarks. The authors should be congratulated for this nice work.

My only concern is that this work does not provide the novelty required to publish in Nature Comm..

For any other physical chemistry journals, I would recommend "Publish as it is" but I cannot recommend publication for this journal.

PS: The NHE scale should avoided and changed to SHE. Indeed, the concept of normality has been abandoned by IUPAC decades ago.

Response to Reviewers' Comments

Reviewer #1

The authors present a carefully executed study on the role of Ag ions as hole-transfer mediator during photoelectrochemical water oxidation. This is a very important study that will influence the behaviour of groups developing, testing and carrying out mechanistic studies of photoelectrodes and photocatalysts. Ag⁺ is a common electron scavenger and here it is clearly shown that in addition to its role as SEA during PC it can also enable improved photocurrents and FE for water oxidation. This has 2 main implications (1) many mechanistic studies need to be carefully re-evaluated and (2) it also offers a simple way to potentially improve the efficiency of photoanodes such as WO₃ which have historically shown low FE's for O₂ production. I strongly recommend publication with only a few points to change/clarify.

Reviewer's Comment 1) In the caption of **Scheme 1** it is stated that band bending is precluded as the WO₃ size is < 100 nm. Has this actually been checked? Depletion widths are typically < 50 nm. This is important as throughout the manuscript the authors state that the applied bias during PEC experiments lowers Ag⁺ reduction and enhances Ag⁺ oxidation by holes. If the WO₃ is fully depleted what is the mechanism by which holes are swept to the surface but electrons moved into the bulk?

Authors' Reply 1) Thank you for this constructive comment. The band diagram in **Scheme 1** was just a simplified illustration for the photoinduced charge transfers occurring on the irradiated WO₃ particles (PC reactions) and particulate WO₃ electrodes (PEC reactions). Accordingly, space charge layers and band bending were not shown for direct comparison of both PC and PEC systems. Regardless of the degree of band bending, however, the valence band edge that belongs to an intrinsic property is unaltered with respect to E°(Ag²⁺/Ag⁺) by applied potential. In addition, the conduction band electrons move to a counter electrode (Pt) and they are not involved with the Ag redox reaction due to separation of the WO₃ and Pt electrode compartments.

Per your comment, **Scheme 1** caption was revised as follows.

Scheme 1. *A simplified band diagram of WO₃ coupled with the electrolyte redox levels and the photoinduced charge transfers occurring on the irradiated WO₃ in the presence of AgNO₃ (pH 5), (a) under open circuit (i.e., bias-free) condition and (b) biased at +0.74 V vs. Ag/AgCl (equivalent to 1.23 V vs. RHE). They are denoted as photocatalytic (PC) and photoelectrocatalytic (PEC) processes, respectively, throughout the text. E_{CB} and E_{VB} represent conduction band edge (+0.03 V) and valence band edge (+2.73 V), respectively. Helmholtz layer potential drop at the interface and the redox energy level broadening caused by thermal fluctuation (i.e., reorganization energy) are not represented. ~~Band bending at the interface is precluded due to WO₃ particle size (<100 nm).~~ The effect of nitrate complexation on the Ag²⁺/Ag⁺ redox potential is not considered. Colored arrows depict the possible charge transfer steps: (ET1) Electron transfer to Ag(I); (ET2) Electron transfer from the CB edge to substrate (FTO); (HT) Hole transfer to Ag(I, II); (RE1) Recombination between electrons in the CB edge and Ag(II, III); (RE2) Recombination between electrons in the FTO and Ag(II, III). Thicker arrows indicate more dominant paths.*

Reviewer's Comment 2) Lines 178-180. A O₂ leak is mentioned and it is stated that a constant value of 0.27 μmol is subtracted off. If O₂ is leaking into the cell I would expect the amount of leaked in air would increase with time. Is this value actually a leak rate e.g. μmol h⁻¹?

Authors' Reply 2) The amount of O₂ (0.27 μmol) in the headspace was an initial and constant level that cannot be removed by Ar purging. To clarify the meaning, the following sentences were modified.

(Lines 178-180) The amount of photogenerated oxygen was subtracted by an initial small background O₂ level (0.27 ± 0.09 μmol) **that cannot be removed by Ar purging. The initial background O₂ level did not increase over 12 h in the dark control condition, which indicates a negligible air leaking.**

Reviewer's Comment 3) Figure 1. The insets are very small and hard to see - suggest making these separate figures. The FE is an important data set (presumably this should also have error bars

Authors' Reply 3) Figure 1a inset with error bars was redrawn to increase visibility.

Reviewer's Comment 4) Figure 1. I can't see error bars on the O₂ levels w/o Ag⁺.

Authors' Reply 4) Error bars were included in **Figure 1**.

Reviewer #2

This paper presents a solid piece of work showing that Ag(II) can be generated in photoelectrocatalytic experiments using WO₃ on an electrode. The work is well carried out and well written. I have no specific criticisms or remarks. The authors should be congratulated for this nice work.

Reviewer's Comment 5) My only concern is that this work does not provide the novelty required to publish in Nature Comm. For any other physical chemistry journals, I would recommend "Publish as it is" but I cannot recommend publication for this journal.

Authors' Reply 5) The novelty of this study has been highlighted several times as stated in cover letter, justification, and manuscript. As the Web of Science reveals over 2,000 articles with the combined keywords of *photo** + *silver* + *oxygen*, Ag(I) has been widely used as a conduction band electron scavenger (*i.e.*, $\text{Ag}^+ + e^- \rightarrow \text{Ag}^0$) in order to facilitate the valence band hole-mediated water oxidation (*i.e.*, $4\text{h}^+ + 2\text{H}_2\text{O} \rightarrow \text{O}_2 + 4\text{H}^+$). However, some concerns on the role of Ag(I) as water oxidation promoter through the formation of high valent silver species (*i.e.*, Ag(II)) have been raised; yet up to date no direct PC/PEC evidence has been reported supporting the claims. ***Here we demonstrate, for the first time, the mediated-PEC oxidation of water and concurrent O₂ evolution by the silver redox cycle (Ag(II/I)) in which the oxidized Ag(II) by valence band hole mediates the O₂ evolution using a biased WO₃ mesoporous electrode under the conditions of a mild acidic nitrate solution (pH 5), zero overpotential (1.23 V vs. RHE), and simulated solar light.*** This study further reveals that the photooxidation of water proceeds through parallel mechanistic paths: a homogeneous one mediated by high valent silver species and a heterogeneous one on WO₃ surface. This phenomenon is found neither in the Ag(I)-free PEC reactions nor in the PC (*i.e.*, bias-free) reactions with Ag(I). ***The present study finds that the role of Ag(I) in water photooxidation is not limited to the commonly accepted role as an electron scavenger and calls for more thorough studies on the effect of Ag(I).***

Reviewer's Comment 6) The NHE scale should be avoided and changed to SHE. Indeed, the concept of normality has been abandoned by IUPAC decades ago.

Authors' Reply 6) The NHE scale was changed to the SHE scale.

REVIEWERS' COMMENTS:

Reviewer #1 (Remarks to the Author):

Thank you for the response and the modified manuscript. I have no further comments and recommend publication.